# Orange-Peel-Derived Nanobiochar for Targeted Cancer Therapy

**DOI:** 10.3390/pharmaceutics14102249

**Published:** 2022-10-21

**Authors:** Daniela Iannazzo, Consuelo Celesti, Claudia Espro, Angelo Ferlazzo, Salvatore V. Giofrè, Mario Scuderi, Silvia Scalese, Bartolo Gabriele, Raffaella Mancuso, Ida Ziccarelli, Giuseppa Visalli, Angela Di Pietro

**Affiliations:** 1Department of Engineering, University of Messina, Contrada Di Dio, 98166 Messina, Italy; 2Department of Clinical and Experimental Medicine, University of Messina, Via Consolare Valeria, 98125 Messina, Italy; 3Department of Chemical, Biological, Pharmaceutical and Environmental Sciences, University of Messina, Viale F. Stagno d’Alcontres, 98166 Messina, Italy; 4Institute for Microelectronics and Microsystems, National Research Council (CNR-IMM), Ottava Strada n.5, 95121 Catania, Italy; 5Laboratory of Industrial and Synthetic Organic Chemistry (LISOC), Department of Chemistry and Chemical Technologies, University of Calabria, Via Pietro Bucci 12/C, 87036 Arcavacata di Rende, Italy; 6Department of Biomedical and Dental Sciences and Morphological and Functional Images, University Hospital of Messina, Via Consolare Valeria, 1, 98100 Messina, Italy

**Keywords:** carbon-based nanomaterials, nanobiochar, targeting ligands, anticancer activity

## Abstract

Cancer-targeted drug delivery systems (DDS) based on carbon nanostructures have shown great promise in cancer therapy due to their ability to selectively recognize specific receptors overexpressed in cancer cells. In this paper, we have explored a green route to synthesize nanobiochar (NBC) endowed with graphene structure from the hydrothermal carbonization (HTC) of orange peels and evaluated the suitability of this nanomaterial as a nanoplatform for cancer therapy. In order to compare the cancer-targeting ability of different widely used targeting ligands (TL), we have conjugated NBC with biotin, riboflavin, folic acid and hyaluronic acid and have tested, in vitro, their biocompatibility and uptake ability towards a human alveolar cancer cell line (A549 cells). The nanosystems which showed the best biological performances—namely, the biotin- and riboflavin- conjugated systems—have been loaded with the poorly water-soluble drug DHF (5,5-dimethyl-6a-phenyl-3-(trimethylsilyl)-6,6a-dihydrofuro[3,2-*b*]furan-2(5*H*)-one) and tested for their anticancer activity. The in vitro biological tests demonstrated the ability of both systems to internalize the drug in A549 cells. In particular, the biotin-functionalized NBC caused cell death percentages to more than double with respect to the drug alone. The reported results also highlight the positive effect of the presence of oxygen-containing functional groups, present on the NBC surface, to improve the water dispersion stability of the DDS and thus make the approach of using this nanomaterial as nanocarrier for poorly water-soluble drugs effective.

## 1. Introduction

Outstanding advances have been achieved in the diagnosis and treatment of cancer diseases over the half past century, enabling new insights into cancer’s underlying molecular mechanisms and in the development of new tools to detect and fight it [1]. These advances and the prevention actions have greatly contributed to the continuous decrease in the age-standardized cancer death rate. However, despite these efforts, cancer still remains a major public health problem worldwide, leading to ever-increasing deaths, globally [2]. The main approaches used today to treat cancer diseases include chemotherapy, radiation therapy, endocrine therapy and immunotherapy, together with surgical interventions [3]. Chemotherapeutics are actually able to kill tumor cells and/or to inhibit their growth and proliferation, but the reported lack of selectivity between cancer cells and normal cells still represents the main reason for therapeutic failure. In fact, the reported toxicity, side effects, and drug resistance continue to be major concerns in cancer therapy [4,5]. Over the past two decades, small molecule-targeted inhibitors have been used as an alternative to broad-spectrum cytotoxic drugs. When compared to traditional chemotherapeutics, these drugs can target cancer cells in a more selective way with respect to normal cells. However, there are still some challenges that these small targeted anti-cancer drugs must face, such as the development of drug resistance [6].

Nanoparticles (NP) technology has opened extraordinary avenues in cancer therapy, allowing the development of cancer-targeted drug delivery systems (DDS) able to improve the efficacy of conventional anticancer drugs [7,8]. The great interest in nano-scaled DDS is due to the appealing features that NP possess. These features include ameliorated drug pharmacokinetics, improved cell uptake, selective cancer targeting, and possible imaging modalities [9]. Smart nanosized DDS, such as polymeric nanoparticles, liposomes, transfersomes micelles, dendrimers, nanoshells, carbon-based nanomaterials, metals and metal oxide nanoparticles have been exploited as suitable platforms for the targeted delivery of anticancer agents, showing an improved safety profile and enhanced anticancer activity when compared with the free cytotoxic drugs [10,11]. The preferential release of therapeutics to cancer cells using NP can be achieved following two ways: “passive” or “active” targeting [9]. Passive targeting is related to anatomical and pathophysiological changes in the cancer vasculature with respect to healthy tissues, which lead to the preferential accumulation of NP at the tumor site over time, a phenomenon known as enhanced permeation and retention (EPR) effect [12]. However, the approved pharmaceutical formulations suffer from limitations mainly related to the lack of control in cell uptake, which often leads to multiple drug resistance [12,13]. The active targeting is based on the introduction, in the NP-based DDS, of targeting molecules able to selectively recognize specific receptors overexpressed on the surface of cancer cells with respect to healthy cells [14,15]. After interactions of the active ligands present on the surface of NP and cancer cells receptors, NP can be internalized inside cells and the cytotoxic drugs released only inside cancerous cells and tissues.

Among the different classes of NP, carbon-based nanomaterials possessing a graphene structure, such as graphene oxide, carbon nanotubes, fullerenes and graphene quantum dots, have shown great potential in cancer diagnosis and treatment [16,17,18,19,20,21]. The presence of many active groups on the surface of these graphene-based nanomaterials (GBM) allows their multimodal conjugation with biocompatible polymers, drugs and targeting ligands (TL), making them ideal nanocarriers for anticancer therapy [22]. In particular, small graphene fragments, such as graphene quantum dots, have been shown to increase the chemotherapy efficacy of anticancer drugs by efficiently accelerating, after endocytosis, the nuclear accumulation of anticancer agents [23]. The cytotoxicity of these nanomaterials was found to be highly dependent on cell uptake, which was greater and delayed using targeting ligand functionalized nanostructure, compared with the free drug alone [24].

Despite the outstanding chemical, physical and biological properties of GBM, not all the members of this family have a good impact on human health and environment. Toxicity and environmental issues are mainly related to the carbon source and methods used for their synthesis, which affect the chemical composition, size and shape of the synthesized nanomaterials [25]. However, new green routes for GBM synthesis have been recently exploited, starting from renewable resources, such as lignocellulosic biomass waste or biocompatible compounds [26,27,28]. Among these routes, the hydrothermal carbonization (HTC) of different biomass wastes allows the graphitization of carbon matrix, producing biochar, without the use of growth catalyst, thus avoiding impurities and additional cleaning procedures [29]. Nanobiochar (NBC), the nanoscale biochar, has received much attention for its significantly higher surface-to-mass ratio than biochar and for its multiple potential biomedical applications [27].

In this work, we have synthesized fluorescent NBC in small size (<100 nm), endowed with graphene structure, by a simple HTC method and exfoliation procedure, starting from orange peel waste [30,31]. Then, in order to assess the possibility to use this nanomaterial as a smart nanocarrier for the targeted cancer cells delivery, different TL have been covalently conjugated to the graphene surface by means of a cleavable and biocompatible polyethylene glycol (PEG) linker [32]. The vitamin-based molecules riboflavin (vitamin B2), biotin (vitamin B7) and folic acid (vitamin B9), as well as hyaluronic acid, have been widely investigated in drug delivery, since their corresponding receptors are overexpressed on the cancer cells’ surface to sustain their rapid growth and proliferation [33]. Accordingly, we have investigated and compared the cancer-targeting abilities of the above-reported four TL by evaluating the uptake of TL-conjugated NBC in the cancer cell line A549, an in vitro model of lung carcinoma. The nanosystems which showed the best biological performances have been loaded with a poorly water-soluble anticancer agent. In particular, the anticancer drug chosen for this study, 5,5-dimethyl-6a-phenyl-3-(trimethylsilyl)-6,6a-dihydrofuro[3,2-*b*]furan-2(5*H*)-one (DHF), is a bicyclic heterocycle, previously synthesized by us, endowed with significant antitumor activity on different cancer cell lines by activating an intrinsic apoptotic mechanism (Figure 1). The results of in vitro biological tests performed on the NBC-based systems have shown the great biocompatibility of the synthesized nanocarriers and the best results, in terms of uptake ability, for the biotin- and the riboflavin-conjugated samples, which, after increased drug-cells interactions, caused cytotoxicity in the investigated cancer cell line. The high water dispersibility of the NBC-based systems has shown the potential to overcome some critical limitations of currently used anticancer drugs.

## 2. Materials and Methods

### 2.1. Materials

All reagents and solvents were purchased from Merck Life Science and used without further purification. Sicilian orange peels (OP) were used in this work as raw material for the production of NBC. DHF was synthesized as previously reported [34].

### 2.2. Chemical, Physical, and Morphological Characterization

Infrared spectra were obtained using a Fourier-Transform Infrared (FT-IR) Spectrum Two FT-IR Spectrometer (PerkinElmer Inc., Waltham, MA, USA) by the ATR method in the range of 4000–500 cm^−1^. Thermogravimetric analyses were carried out at 10 °C/min, from 100 to 1000 °C, in an argon atmosphere using a TGA Q500 instrument (TA Instruments, New Castle, DE, USA). The particle size and zeta potential measurements were performed by dynamic light-scattering (DLS) analyses using the Zetasizer 3000 instrument (Malvern, Worcestershire, England), equipped with a 632 nm HeNe laser, operating at a 173-degree detector angle. Micro-Raman spectra were recorded in backscattering geometry using a LabRam HR 800 spectrometer (Horiba, Ltd., Kyoto, Japan). UV spectra have been performed by Thermo Nicolet mod, Evolution 500 spectrophotometer. X-ray powder diffraction (XRD) patters were carried out on a Bruker AXS D8 Advance X-ray diffractometer, using the CuKa1 radiation. The NBC morphology was investigated by transmission electron microscopy (TEM); an aqueous solution containing NBC was dropped out on a lacey-carbon TEM grid. TEM analyses were performed in a probe aberration-corrected JEOL JEM-ARM200F microscope (JEOL USA, Inc., Peabody, MA, USA), operated at the primary beam energy of 200 keV. Photoluminescence (PL) analyses were performed using a spectrofluorometer NanoLog modular (Horiba, Ltd., Kyoto, Japan) under excitation with a xenon lamp; the nanomaterials’ water dispersions were analyzed at the concentration of 100 ng/mL exciting the samples at the excitation wavelengths from 320 nm to 360 nm. Confocal Laser Scanning Microscopy (CLSM) images were performed by a confocal microscope equipped with a 40× 1.0 NA immersion objective and TCS SP2 instrument (Leica Microsystem Heidelberg, Mannheim, Germany).

### 2.3. Synthesis of NBC-Based DDS

The nanosystems used in this study have been produced by HTC treatment of orange peels and subsequent organic functionalization of the synthesized nanomaterials with different TL and DHF, following the synthetic protocol reported below and schematized in Figure 2.

#### 2.3.1. Synthesis of NBC

The general preparation procedure of hydrochar via HTC of orange peels was performed as previously reported [30,31]. Briefly, in a 300 mL stainless-steel autoclave under autogenous pressure and air atmosphere, 20 g of OP, reduced in small pieces, and 50 mL of deionized water were added, and the reactor was heated at 240 °C for 1 h at a stirring speed of 600 rpm. Then, the solid product was separated by vacuum filtration with a Buchner funnel and filter paper, and the solid hydrochar was washed with warm distilled water and dried overnight at 60 °C, under vacuum. The obtained hydrochar was treated with a water solution of NaOH 2 M, and the suspension was filtered under vacuum through a 0.1 µm Millipore membrane. The filtrate was added with HNO_3_ solution until a neutral pH was achieved and purified using dialysis bags (12,000 Dalton molecular weight). A small amount of the resulting dispersion was dried at 60 °C, under vacuum, and used for the FTIR, Raman and TGA characterizations. The number of acidic groups present on the NBC surface was evaluated in water dispersions as a function of zeta potential using the instrument Zetasizer 3000 (Malvern) in the pH value range of 6–8 by employing, as titrants, solutions of 0.1 M HCl or 0.1 M NaOH and was found to be of 2.26 mmol/g.

#### 2.3.2. Synthesis of NBC-PEG

A dispersion of NBC (20 mg) in dimethylformamide (DMF, 10 mL) was treated with N-(3-dimethylaminopropyl)-N′-ethylcarbodiimide hydrochloride (EDC·HCl, 10 mg, 0.052 mmol) and triethylamine (ETA, 7.2 µL, 0.052 mmol), and the mixture was stirred for 15 min at room temperature. Then, the mixture was added with hydroxybenzotriazole (HOBt, 7.0 mg, 0.052 mmol) and a catalytic amount of 4-dimethylaminopyridine (DMAP) and left to stir for an additional 1 h. The dispersion was treated with 33.6 mg of O-(2-aminoethyl)-O′-[2-(Boc-amino)ethyl]decaethylene glycol (0.052 mmol), and the reaction mixture was left under stirring at room temperature for 4 days. The obtained dispersion was diluted with deionized water and then purified using dialysis bags (MW of 12,000 Da) for 4 days. The Boc-protecting group was removed by treating the sample with HCl (4 M) in dioxane at room temperature for 1 h at room temperature. The NH_2_ loading of the obtained NBC-PEG sample, as calculated by a Kaiser test, was found to be of 1.2 mmol/g.

#### 2.3.3. Synthesis of NBC-TL Samples

Solutions of 5 mg of biotin (B, 0.02 mmol), folic acid (FA, 0.01 mmol) or hyaluronic acid (HA, MW = 20–40 kDa) in DMF (10 mL) were treated with equimolar amounts of EDC·HCl and ETA, and the solutions were left to stir at room temperature for 15 min. Then, an equimolar amount of HOBt and a catalytic amount of DMAP were added, and the resulting mixture was left to stir for 1 h. For the NBC-R system containing Riboflavin (R) as targeting ligand, the reaction with succinic anhydride was needed to introduce the carboxyl group needed for the subsequent coupling reaction with the nanosystem. Briefly, following the above-reported synthetic protocol, 5 mg of riboflavin (0.013 mmol) in DMF (10 mL) was treated with two equivalents of EDC·HCl, ETA, HOBt, a catalytic amount of DMAP and an equimolar amount of succinic anhydride, and the resulting mixture was left to stir for an additional 1 h. An NBC-PEG (5 mg) sample was then added to the suspensions of each activated TL sample, which were left under stirring for 5 days at room temperature. The obtained dispersion was diluted with deionized water and purified for 4 days, using dialysis bags (MW of 12,000 Da).

### 2.4. Synthesis of DHF@NBC-B and DHF@NBC-R Samples

A dispersion of 5 mg of NBC-B or NBC-R samples in phosphate buffer solution (PBS) at pH 7.4 (5 mL) was treated with a solution of DHF (5 mg) in 5 mL of PBS at room temperature for 5 days. The nanosystems were then purified using dialysis bags of MW 12,000 Da for 4 days, until no amount of unbound DHF was detected in the washing solutions. Known amounts of the resulting materials were dried at 60 °C, under vacuum, for further characterizations.

### 2.5. Biological Studies

Sub-confluent monolayers (75%) of A549 cells (human alveolar cell line: ATCC-CCL-185Tm) were used as a biological model to assess the biocompatibility and the uptake of NBC as well as the cytotoxicity exerted by the drug-loaded NBC in comparison to the free drug at the same concentrations. Briefly, cells were cultured in RPMI 1640 medium (Merck Life Science S.r.l.) supplemented with 2 mM of L-glutamine, 10% of inactivated fetal bovine serum (FBS) and 1% penicillin/streptomycin at 37 °C in a 5% CO_2_/95% air humidified atmosphere. For the experiments, A549 cells were cultured for 24 h in 96-well microplates (final density 4 × 10^4^ cells/well) and treated for 24 h with the sample suspensions prepared in cell medium (with 2% FBS). The uptake measurements of TL-functionalized NBC in A549 were performed by spectrofluorimetric analyses by using a microplate reader (Tecan Italia, Milan, Italy). After abiotic experiments for the selection of wavelengths of excitation and emission of all samples in the range of 12.5–200 µg mL^−1^, we observed a strong positive correlation between emission values and sample doses (Pearson correlation coefficient [r] > 0.95; *p* < 0.01) and, adjusting the gain, the λ_exc_ of 485 nm and λ_em_. of 535 nm were selected for all samples. It is noteworthy that, with the same gain, the NBC-FA had a 19.3 times lower emission than NBC-B, which in turn had an emission 1.4 times lower compared to NBC-R. Finally, even by increasing the gain to very high values, the emission values of NBC-HA were negligible and, in comparison to the non-functionalized NBC, they were more than halved. This did not allow us to evaluate, by spectrofluorimetric analyses, the uptake of NBC-HA in A549 cells. After treatment of A549 for 24 h with the sample suspensions at doses of 50 and 100 µg mL^−1^ (8 wells for each sample), fluorescence reading was carried out and emission values were recorded before removing the medium. Then, the monolayer was washed repeatedly with PBS, and emission values were recorded to measure the percent uptake of NBC-TL, which was confirmed by confocal microscopy observations. The assessment of biocompatibility of the NBC-TL and the drug-induced cell death of DHF@NBC-TL was performed by colorimetric MTT assay, based on the reduction of 3,(4,5-dimethiazol-2)-2,5-difeniltetrazolium bromide), catalyzed by cellular NAD(P)H-dependent dehydrogenases. Briefly, the assay was performed in cells that were cultured for 24 h in 96-well microplates, to which the appropriate volume of the samples was added in maintenance medium (with 2% FBS). After 24 h, 0.5 mg mL^−1^ of MTT (100 μL) in RPMI without phenol red (Sigma, Milan, Italy) was added to each well, and the microplates were incubated at 37 °C for 3 h to allow metabolic activity. By using a mixture containing 50 mM HEPES (pH 8.0) and ethanol (1:9, *v*/*v*) to solubilize the formazan crystals, the enzymatic activity was quantified by spectrophotometric measurement at 540 nm using a microplate reader (Tecan Italia). The optical density (OD) values obtained for each sample were compared with the mean OD of the negative control (i.e., PBS treated cells), which was arbitrarily considered corresponding to 100% viability. Free drug at the same doses present in NBC were used as positive controls, using a stock solution (10 mg/mL in DMSO). This allowed us to nullify the cytotoxicity of DMSO, which was nontoxic even for the highest doses of drug tested. All biological tests were performed in triplicate and the replicates were calculated as means ± SD.

## 3. Results and Discussion

### 3.1. Nanobiochar Preparation

The nanostructured material to be used as biocompatible nanocarrier for targeted cancer therapy was synthesized by hydrothermal carbonization (HTC) of orange peels at 240 °C for 1 h. This green top-down synthetic method allows the graphitization of the carbon matrix starting from a natural renewable source, without the use of growth catalyst, thus avoiding impurities and additional cleaning procedures [30,31]. The synthesized carbonaceous material, obtained as previously reported by a series of deoxygenating processes from cellulose, hemicellulose and lignin, with 29% yield, has been shown to contain many oxygenated functionalities on the graphitized surface [30,31]. The biochar was treated with a NaOH 2 M solution and filtered through a 100 nm membrane filter. The filtrate solution was added with a dilute solution of HNO_3_ until a neutral pH was achieved, and the nanomaterial was purified by a dialysis-bag technique in order to avoid nanoparticle aggregation, until no organic or inorganic materials were recovered in the water washing solutions, to finally give nanobiochar (NBC). The morphology of the nanomaterial was investigated by TEM analyses, which showed the presence of nanoparticles with dimensions ranging from 10 to 100 nm in diameter. The representative images of NBC structures are shown in Figure 3a, where two nanostructures of different sizes (roughly 60 nm and 20 nm) are reported. Each single nanostructure is made by a multilayered graphene arranged in concentric manner around a hollow center, like a multilayer graphene shell around a hollow core. In Figure 3b a small nanostructure of 10 nm with a very small hollow core is shown. The high-resolution TEM micrograph in Figure 3c shows an enlarged in-plane view of the two nanostructures present in Figure 3a. The multilayer graphene structures have different thicknesses—namely, 30 and 10 graphene layers for the big and small nanostructure, respectively—with an interlayer spacing d002 of 3.4 Å, which is normally found in graphene-like structures (line-scan on the inset of Figure 3c) [35].

DLS measurements allowed us to confirm the small size of the synthesized nanomaterial as well as its water dispersion ability (Figure 4a) The volume-weighted DLS measurements revealed a single population centered at 67.4 nm with a polydispersity index (PDI) value of 0.2, which further confirms the tight particle-size distribution. The zeta potential values of the nanomaterials, evaluated in deionized water at the pH range of 6.0–8.0, were found to be always lower than −30 mV (Figure 4b), thus validating the high dispersion stability in water [36]. This dispersibility can be related to the presence of acidic functionalities present on the carbon surface, which, as evaluated by titration analysis, were found to be of 2.26 mmol/g. This value is very similar to that previously obtained from graphene quantum dots synthesized by acidic oxidation of multiwalled carbon nanotubes (2.37 mmol/g), thus validating the effectiveness of our green synthetic method [32].

The chemical structure of the synthesized nanomaterial was also investigated by Raman, XRD and FTIR measurements, whereas the optical properties have been evaluated by PL analysis (Figure 5). The Raman spectra of NBC show the distinctive D-band at 1352 cm^−1^ and the G-band at 1580 cm^−1^ normally encountered for sp^2^ carbon nanomaterials (Figure 5a). The intensity ID/IG ratio, which was found to be of 0.74, indicates a high graphitization degree; this value is lower than that reported from other carbon-based nanomaterials obtained from other biomass sources [37,38]. The XRD diffraction spectrum of NBC (Figure 5b) shows a broad (002) diffraction peak centered at 2θ = 18.5° related to the interlayer graphene spacing, which is similar to that obtained from small graphene-based materials obtained by other methods [39,40]. The FTIR spectrum of NBC (Figure 5c) clearly demonstrates the presence of oxygenated groups on the nanomaterial surface. In particular, the broad band centered at 3410 cm^−1^ can be ascribed to the stretching of the O−H bond, whereas the peaks at 1368 cm^−1^ and at 1019 cm^−1^ are attributable to the O–H bending of phenol and alcohol groups, respectively. Moreover, the presence of a peak at 1710 cm^−1^ due to the vibrations of the C=O bond of the carboxyl functionality and a peak at 1619 cm^−1^ related to C=C stretching vibrations of alkenyl functionalities can be observed.

The small dimension of NBC was also demonstrated by evaluating the optical properties of the synthesized nanomaterials (Figure 5d). The UV−vis spectrum of NBC shows the absorption band at ∼250 nm corresponding to the π−π* transition of sp^2^ aromatic domains. Photoluminescence measurements confirm the emission properties of the nanomaterial since, when exciting the NBC water dispersion at the excitation wavelengths from 320 to 360 nm, a strong peak at 560 nm can always be observed.

### 3.2. Targeting Ligands Functionalized NBC

The synthesized NBC was investigated for its ability to selectively deliver anticancer agents into tumor cells and, in general, as a nanocarrier to evaluate the best targeting ability towards A549 cells of different targeting ligands (TL). In particular, the TL riboflavin (R, vitamin B2), biotin (B, vitamin B7), folic acid (FA, vitamin B9) and hyaluronic acid (HA), widely used in studies related to cancer-targeted therapy, have been covalently linked to the graphene surface by means of a cleavable and biocompatible PEG linker.

#### 3.2.1. Synthesis of NBC-TL Samples

The synthetic strategy towards the synthesis of TL-conjugated NBC involved the reaction of the carboxyl groups present in the graphene surface with the bidentate PEG linker, which is protected at a single amine functionality with the *tert*-butyloxycarbonyl (BOC) group. The coupling reaction was performed in DMF at room temperature using *N*-(3-dimethylaminopropyl)-*N’*-ethylcarbodiimide hydrochloride (EDC·HCl) and hydroxybenzotriazole (HOBt) as coupling agents and in the presence of a catalytic amount of 4-dimethylaminopyridine (DMAP). The subsequent BOC-deprotection afforded the amino-functionalized nanosystem that was then conjugated with the TL possessing a free carboxyl functionality, following the same synthetic strategy. For riboflavin, the preventive coupling reaction with succinic anhydride was performed to introduce the carboxyl group needed for the subsequent coupling reaction with the nanosystem (Figure 1).

After purification by dialysis-bag technique, the TL-conjugated samples were characterized by FTIR, TGA, PL and DLS analyses (Figure 6 and Figure 7). The FTIR analyses of the TL-conjugated samples compared with the NBC-PEG precursor are shown in Figure 6a. All samples show a broad band at around 3400–3600 cm^−1^, due to the presence of O−H bonds and N–H bonds, a peak at around 2930 cm^−1^ due to C–H stretching of PEG chain and a peak at 1020 cm^−1^ ascribable to the C–O alkoxy groups present on the nanosystems after PEG conjugation. The FTIR spectrum of the NBC-PEG sample shows the representative peak at 1660 cm^−1^, related to the C=O stretching of the newly formed amide bond between PEG and NBC and a broad peak at 1632 cm^−1^ due to the N–H bending of amine group. The NBC-B system shows the presence of a peak at 1658 cm^−1^ ascribable to the C=O stretching of the newly formed amide functionality and a broad band at 1450 cm^−1^, due to the N–H stretching. For the NBC-HA system, the additional peak at 1724 cm^−1^ due to the free carboxylic group, together with the representative peak at 1655 cm^−1^ due to the newly formed amide bond, can be observed. The NBC-FA system also shows the presence of a peak at 1665 cm^−1^ due to the C=O stretching of the amide bond together with a sharp peak at 1608 cm^−1^ due to the N–H bending of amide functionality. Finally, the NBC-R system shows the presence of two distinctive peaks at 1710 cm^−1^ and 1634 cm^−1^ due to the C=O stretching of ester functionality and to the N–H bending of the amide group, respectively. The TGA analyses of the TL-conjugated NBC samples, performed under inert atmosphere, demonstrated the thermal stability of the synthesized nanomaterials, since a gradual decomposition is observed for all the investigated samples in the temperature range of 100 to 1000 °C. Moreover, this analysis allowed us to evaluate the degree of functionalization of the NBC-TL samples with respect to the precursor NBC-PEG. For each conjugated system, an increase of weight loss can be observed, whose amounts calculated as difference with respect to NBC-PEG at 1000 °C under inert atmosphere were found to be of 13.24 wt% for NBC-B, 28.68 wt% for NBC-HA, 35.44 wt% for NBC-R and of 42.89 wt% for NBC-R. The observed increases in weight losses, together with the different shapes of TGA profiles of TL-NBC samples, suggest that a deep chemical modification occurred on the nanomaterial after the functionalization processes (Figure 6b).

The surface functionalization of carbon-based nanomaterials can play an important role in the possibility to form self-aggregates or to modify their optical properties [32,41]. Given the importance that these aspects play in drug delivery, we investigated the dimensions of the TL-conjugated systems by DLS analyses and their optical properties by PL measurements (Figure 7). The results of volume-weighted size distribution show, for all the samples, single-size populations with dimensions always less than 100 nm (Figure 7a). In particular, we found size populations centered at 50.75 nm for NBC-R (PDI = 0.1), 51.20 nm for NBC-FA (PDI = 0.4), 58.77 nm for NBC-B (PDI = 0.5) and 78.92 nm for NBC-HA (PDI = 0.2). The greater size population observed for NBC-HA could be explained by the tendency of HA conjugates to form nanosized self-aggregates [42]. However, the PDI values (always lower or equal to 0.5) are indicative of the greater water stability of the synthesized nanosystems. The optical properties and the presence of stable interactions between the TLs and NBC were also evaluated by measuring the fluorescence properties of the conjugated samples (Figure 7b). All samples have been shown to maintain the PL properties of the starting NBC at the excitation wavelength of 360 nm. Only for the NBC-HA sample was a decrease in PL intensity recorded. This decrease can be ascribed to the greater size of the nanoparticles after HA promoted aggregation processes, leading to the formation of self-assembled aggregates [43]. The recorded blueshift of the conjugated samples from NBC (PL maximum intensity at 560 nm) to 539 nm, 542 nm, 531 nm and 528 nm for NBC-R, NBC-B, NBC-FA and NBC-HA samples, respectively, are indicative of the presence of a stable interaction between NBC and TL [32].

#### 3.2.2. Cellular Uptake and Biocompatibility of TL-Functionalized NBC

The ability of the TL-functionalized NBC to deliver drugs to cancer cells was tested against A549 cells, after evaluating the uptake and biocompatibility of the nanosystems. As previously discussed, the spectrofluorimetric analyses did not allow us to evaluate the uptake of the NBC-HA. This result is in complete agreement with the observed PL behavior of this sample (see Figure 7b) and can be rationalized by the improved nanoparticles’ aggregation (see Figure 7a), attributable to hydrogen bonding interactions between the carboxylic groups present in HA and the oxygen functionalities exposed on the NBC surface, which lead to quenching phenomena.

The uptake measurements of NBC-FA, NBC-B and NBC-R at the doses of 50 and 100 µg mL^−1^, in our cell model, are reported in Figure 8, which also reports the values obtained in cells treated with pegylated NBC, without any TL. The results of fluorometric analyses highlighted particularly high values of uptake for NBC-B and NBC-R, since, in comparison to the total emission values, those recorded in the monolayers after removal of the noninternalized NBC-TL (i.e., suspended in the medium) were always ≥75%. Conversely, A549 cells showed a lower uptake ability for NBC-FA, whose percentage at the highest dose tested was of 25, showing values increased by only 50% compared to pegylated NBC. The uptake of NBC-B and NBC-R was confirmed by confocal microscopy observations that showed the intracellular presence of biotin-conjugated NBC (Appendix A, ESI). Despite the low emission of biotin, fluorescent dots were mainly present in the perinuclear area. Based on the results of cellular uptake, biocompatibility was evaluated in the dose interval 40–200 µg mL^−1^ exclusively on NBC-R and NBC-B, which showed low cytotoxicity. The observed low cytotoxicity of nanosystems further highlights the advantage of using graphene-based materials obtained from natural sources instead of more traditional sources [44]. In comparison to the untreated control cells, the cell viability rates, expressed as the percentage of viable cells, at the highest tested dose were 88.3 (±5.9) and 83.5 (±7.4) for NBC-B and NBC-R, respectively. These data were further confirmed by microscopic observations, which highlighted that the NBC-TL treatment did not change the typical epithelial morphology of A549.

### 3.3. NBC-Based DDS

#### 3.3.1. Synthesis of DHF@NBC-R and DHF@NBC-B

The ability to selectively deliver anticancer agents to tumor cells was tested for the TL-conjugated nanocarriers that showed the best uptake ability towards the selected cancer cells—namely, NBC-B and NBC-R. The model drug chosen in this study was a bicyclic heterocycle, previously synthesized by us [34], which showed good cytotoxicity towards cancer cells by activating an apoptotic mechanism (Figure 9). The loading of DHF was performed in basic buffer solution at pH 7.4 in order to allow the effective π−π stacking and hydrogen bond interaction with the graphene, as reported in the literature for similar systems [45]. The stable but not covalent interaction of this drug with the nanocarrier can facilitate its prompt release inside cells, after endocytosis. The aromatic ring of DHF can allow the effective π−π stacking interaction with the graphene surface, whereas the carbonyl group can interact with the hydroxyl moieties present in the nanomaterial by hydrogen bonding (Figure 9).

The amounts of drug loaded on the nanosystems, as evaluated by percentage of weight losses at 500 °C under inert atmosphere with respect to the precursors, were found to be of 25 wt% for DHF@NBC-B and 4.1 wt% for DHF@NBC-R. The lower amount of DHF loaded on the NBC-R system with respect to the biotin-conjugated one can be explained by the steric hindrance of riboflavin and its stable interaction with the graphene surface, which prevents a large interaction of the drug with the nanomaterial, as also reported in similar studies [46,47]. The presence of a stable interaction of DHF was investigated by FTIR spectroscopy and by PL measurements. The FTIR spectra of the drug-conjugated samples DHF@NBC-B and DHF@NBC-R show the presence of the characteristic peaks present in the drug. In particular, the stretching of the C=O ester group at 1720 cm^−1^ together with the Si–CH_3_ stretching vibration peak at 1250 cm^−1^ are clearly evident [48] (Figure 10a). The stable interaction of DHF with the nanosystems was also proved by PL spectra by comparing the fluorescence properties of NBC-R and NBC-B with those of the drug-conjugated samples at the excitation wavelength of 360 nm (Figure 10b). The PL spectra clearly show quenching phenomena, which can be ascribed to the electron transfer process triggered by π–π stacking interactions between the phenyl ring of DHF and the p-cloud of NBC. This decrease in the fluorescence intensity together with the blueshift of conjugated samples from 539 nm (in the NBC-R sample) to 527 in the DHF@NBC-R and from 542 nm (in the NBC-B sample) to 532 in the DHF@NBC-B system, respectively, are indicative of the stable chemical interaction between the drug and the nanocarriers [30].

In order to evaluate the effect of the NBC surface functionalization on the water dispersibility, we have calculated the electrophoretic mobility of the drug-conjugated systems in deionized water (Figure 11). The DLS analyses showed for both DDS zeta potential values lower than −30 mV (−32.66 mV for DHF@NBC-R and −46 mV for DHF@NBC-B). The obtained values further confirm the high stability of the synthesized nanosystems in water-based solvents. Moreover, we have sonicated the drug-conjugated samples DHF@NBC-B and DHF@NBC-R in deionized water by dispersing 2 mg of each sample in 10 mL of water. As shown in the inset of Figure 11a,b, a good water dispersion stability can be observed for both samples after 2 days. These results highlight the positive effect of the presence of carboxylic functionalities on the water dispersion stability of NBC and the effective approach in using NBC as DDS for poorly water-soluble drugs, such as DHF.

#### 3.3.2. Evaluation of Drug-Induced Cell Death of DHF@NBC-TL

To assess the anticancer activity of the drug-conjugated samples DHF@NBC-B and DHF@NBC-R, we measured the cytotoxicity of the nanosystems and compared the reported activities with that exerted by the anticancer drug alone at the same doses. Figure 12 shows the results obtained in A549 cancer cells treated for 24 h with DHF@NBC-B and expressed as percentages of dead cells (treated/untreated cultures). Considering the above-reported amounts of drug in the DHF@NBC-B, the tested drug doses were 10, 25 and 50 µg/mL^−1^, which caused cell death percentages, on average, more than double those of the drug alone. Based on these results, the IC_50_ value of DHF, whose drug loading in the DDS was 25 wt%, was found to be of 35.65 µg mL^−1^ in DHF@NBC-B vs. 103.2 µg mL^−1^ for the drug alone, thus highlighting the significantly high ability (*p* < 0.01) of the biotin-functionalized NBC to internalize the drug in these cancer cells. A plausible mechanism of internalization of the DHF@NBC-B could be explained through the biotin-mediated interaction between DDS and cell membrane followed by the endocytosis of the DDS. In the same cell model, this internalization was previously observed by using similar carbon-based nanosized delivery systems [25,30]. The low amount of DHF loaded on the NBC-R system (4.1 wt%) did not allow us to compare higher drug concentrations, except by using very high concentrations of the DDS. However, in comparison to DHF alone, the drug dose of 10 µg/mL^−1^ caused a percentage of cell death equal to 17.62 (±1.9) vs. 9.63 (±1.1) of DHF alone. This result confirmed the high cell uptake and anticancer activity of this delivery nanosystem. On the other hand, the lowest drug loading could limit the possibility of this system to deliver DHF. Different functionalization methodologies can be further exploited for this latter system to overcome the encountered drug delivery limitations.

## 4. Conclusions

We have reported the synthesis of a nanostructured graphene-based material by hydrothermal carbonization of orange peels as a renewable biomass source. This green, top-down synthetic route easily led to a fluorescent and highly water-dispersible nanomaterial endowed with graphene structure, without the use of growth catalysts. The synthesized NBC was investigated as nanocarrier for the targeted delivery of drugs to cancer cells. In order to evaluate and compare the cancer-targeting ability of different cancer-targeting molecules, we have covalently conjugated NBC with biotin, riboflavin, folic acid and hyaluronic acid by coupling reactions at room temperature. The evaluation of their biocompatibility and uptake ability towards the A549 cancer cell line showed the best biological performances for biotin and riboflavin conjugates, which were loaded with the poorly water-soluble drug DHF. The in vitro evaluation of anticancer activity against the selected cancer cell line demonstrated the ability of biotin- and riboflavin-conjugated nanosystems to internalize drugs in cancer cells, probably by receptor-mediated endocytosis, causing cell death. In particular, the biotin-functionalized NBC showed cell death percentages more than double with respect to the drug alone.

The results of this study underline the positive effect of the oxygen-containing functional groups, present on the NBC surface after simple heat treatment of a natural source, on the water dispersion stability of NBC, allowing the possibility to incorporate anticancer drugs poorly soluble in water.

The demonstrated suitability of the NBC system to be used as nanocarrier for drug delivery and the possibility offered by the multi-functionalization of graphene-based materials may open the way to the development of new NBC-based nanocarriers able to minimize the anticancer drugs’ systemic toxicity and undesirable side effects typically associated with conventional chemotherapy. The investigated nanocarrier is extremely versatile, as the drug could be used for multiple cancers simply by changing the target ligand. Further biological studies performed on breast, prostate and cervical cancer cells with vitamin receptors that will preliminarily be quantified could, of course, lead to a better understanding of the drug internalization mechanisms as well as the development of more cancer-targeted formulations.

## Data Availability

Not applicable.

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
