# Peer review of "Orange-Peel-Derived Nanobiochar for Targeted Cancer Therapy"

_pharmaceutics, 2022, doi:10.3390/pharmaceutics14102249_

Round 1
Reviewer 1 Report (New Reviewer)
The manuscript "Orange peel derived nanobiochar for targeted cancer therapy " describes the green-synthesis of graphen-like drug delivery system from orange peels, followed by endowing the system with a number of cancer cell targeting molecules and using the most efficient system to deliver a hydrophobic cytotoxic drug designed by authors to A549 cancer cell line. The green synthesis of nanoparticles from waste is a very perspective approach because of its cheapness and sustainability. Thus, the topic is actual, the experimental design is logical, the study is novel and fully within the scope of Pharmaceutics. However, some comments need to be addressed:
1. Line 192: please decipher the abbreviation NBC-R (and other abbreviated drug delivery systems )at first appearance in the text.
2. Figure 12. The structure of synthesized nanobiochar was similar to those of graphen and carbon nanotubes. The toxicity of graphen and carbon nanotubes is often described (for example: https://doi.org/10.1016/j.clay.2021.106041). Could the toxic effects of DHF@NBC-B be related to the drug delivery systems themselves? The control of cells treated with empty drug delivery system (without drug) is needed.
3. Line 252: What solvents were used when free drug or NBC-drug were added to cells?
4. Could the authors provide additional evidence that NBC were internalized and not just strongly adsorbed on the cell surface?
5. Lines 171-173: Could the authors please briefly describe the measurements details and calculations of the number of acidic groups present on the NBC surface (in the Materials and methods section or as the Supplemental Materials)
6. Lines 408-411: I would call a shift from 560 nm to lower wavelengths of 539 nm, 542 nm, 531 nm and 528 nm a blueshift rather than a redshift. The same is true for the shift from 539 nm (in the NBC-R sample) to 527 in the DHF@NBC-R and from 542 nm (in the NBC-B sample) to 532 nm in the DHF@NBC-B system (lines 480-482)
7. Line 88: Do the authors mean "depended on" cell uptake?
Author Response
Please, see the attachment

Reviewer 2 Report (New Reviewer)
Dear Authors,
this is an interesting paper, but some revisions are need for its publication.
Introduction
-line 81: Ref 16-19 Please add more recent papers for example doi: 10.3390/cancers13091991; doi: 10.3390/ijms222111783
-line 97 ref (24,25): Please add more recent ref for example doi: 10.3390/nano12050885.
Material and methods
-line 152: please add the dot
-line 224 (with 2% FBS...): Why 2% FBS? Why was the basal medium without supplements not used?
-line 225 "uptake measurements of TL function- 224 alized NBC in A549 were performed by spectrofluorimetric analyses": I don't think this is a good way to evaluate uptake, I suggest adding in-cell imaging
-line 240 "cell mortality": replace with cell death
-line 243 "were cultured for 24 h": Cytotoxicity must be performed for at least 72 hours. Add the times 48 and 72h
Results
-line 339 "PEG": check that the acronym is already specified
-line 345 "DMF": check that the acronym is already specified
-lines 427-436: see my previous suggestion for line 225
Author Response
Please, see the attachment

Round 2
Reviewer 1 Report (New Reviewer)
The authors have significantly improved the manuscript. In my opinion, it can now be published in Pharmaceutics.
Reviewer 2 Report (New Reviewer)
Dear Authors, in my opinion the revised version of the manuscript is now acceptable for its publication.Best regards
This manuscript is a resubmission of an earlier submission. The following is a list of the peer review reports and author responses from that submission.
Round 1
Reviewer 1 Report
The article entitled Orange peel derived nanobiochar for targeted cancer therapy is a document of interesting subject matter.
However, it needs some major changes before being accepted. Make the following corrections:
1. Authors should add a schematic regarding preparation of nano-sized Transfersomes including drug, for clarity.
2. Conclusion needs to be expanded mentioning the application of the methods developed and use- further future extension of the work.
3. Some of results lack stastical significance. Authors are advised to mention.
4. Why the volume-weighted DLS measurements and not based on Intensity on Number?
5. Please try to explain sharp difference in the size by the DLS and TEM.
6. The authors should cite and discuss some related studies about these nano-sized Transfersomes especially in the cellular uptake.
7. The conclusion is a bit too concise. Please make a general conclusion of the study.
8. Please try to compare the results of your paper with another similar study.
Author Response
Please, see the attachment.

Reviewer 2 Report
The authors Daniela et al. describe a novel route to synthesize a targeting ligand conjugated nanobiochar by hydrothermal carbonization of orange peels for targeted cancer therapy. The manuscript is very well written and easy to follow, the results are clearly presented, and the experimental data are fully described. I have no major concerns about the publication of this manuscript.
A few minor spelling mistakes need to be corrected in some places.
a) What was the loading ratio of NBC: Drug? Describe the cytotoxicity effect of NBC and what concentration it causes cytotoxicity.
b) What parameters could be used to increase the loading amount. (Line 513)
c) As the authors mentioned internalization in the discussion, please suggest a mechanism of internalization of NBC.
Author Response
Please, see the attachment.

Reviewer 3 Report
This article presents a study on the synthesis, property characterizations, and biomedical applications of nanobichar using a biomass waste. In general, the precursor is interesting but, unfortunately, the authors didn't link the content of orange peel to the selection of precursor. In addition, the article cannot be accepted due to many writing and grammar errors. Furthermore, many experimental designs and result discussions lack rigor. For example, the claim of the effect of oxygen-containing groups on the water dispersity of NBC doesn't make sense if without controls. Below are some comments for the authors to improve the manuscript.
1. The authors mentioned twice PL in the section of Chemical, Physical, and Morphological Characterization. Please combine them into one and rewrite the last sentence.
2. I am very confused about the synthesis of NBC. Can NBC dissolve in water or no? Please explain a bit the rationale of each purification step.
3. You can combine 2.5.1, 2, 3, and 4 with highlights on the differences and delete 2.5.5 since you have already introduced it in 2.5.1.
4. When you further doped DHF, was it excessive? Otherwise, with the same MWCO of dialysis bag, undoped NBC will also remain. What is DHF short for?
5. TEM data are not representative. Is there a size distribution histogram?
6. Did you conduct PL spectra of NBC with other excitation wavelengths?
7. Please rewrite this sentence "The synthetic strategy towards the synthesis of TL conjugated NBC involved the reaction of the carboxyl group present in the graphene surface with the bidentate PEG linker protected with the tert-butyloxycarbonyl protecting (BOC) group, at a single amine functionality.", especially in the end. Why at a single amine functionality?
8. Many FTIR peaks in NBC-PEG have been shown in NBC so the authors cannot claim them from PEG.
9. The increase of mass loss in TGA is insufficiently related to the surface functionalization of NBC.
10. In terms of zeta potential measurements, the authors claimed both good stability and self-aggregation in NBC-HA, which contradicts.
11. A PL decrease can be due to energy transfer and is not directly related to self-aggregation.
12. Why did the authors compare different NBC-TL complexes and claimed stable interactions between NBC and all TL? You need to keep with one complex and modify the loading amount or other parameters.
13. Please provide a reference to this explanation "...can be explained by the steric hindrance of riboflavin and on its stable interaction with the graphene surface that forbids a great interaction of the drug with the nanomaterial."
14. Please rewrite "...10, 25 e 50 507 μg/mL-1...".
15. Cell death not cell dead..
Author Response
Please, see the attachment.

Round 2
Reviewer 1 Report
Accept in the current form.